# Spatio-Temporal Pattern and Driving Force Evolution of Cultivated Land Occupied by Urban Expansion in the Chengdu Metropolitan Area

**Bao Meng [1], Xuxi Wang [2], Zhifeng Zhang [3] and Pei Huang [4],***

[1] Faculty of Economics and Business Administration, Yibin University, Yibin 644000, China
[2] Sichuan Institute of Urban and Rural Construction, Chengdu 610000, China
[3] The Faculty Geography Resource Sciences, Sichuan Normal University, Chengdu 610100, China
[4] Institute of International Rivers & Eco-Security, Yunnan University, Kunming 650500, China
[*] Correspondence: hphyyy09@mail.ynu.edu.cn

**Abstract:** Cultivated land is the principal land source for urban expansion. Recent large-scale urban expansion through the occupation of cultivated land has influenced regional food security and the realization of sustainable development goals. Based on data regarding the cultivated land occupied for urban construction in the Chengdu metropolitan area from 2000 to 2018, the spatio-temporal evolution of cultivated land occupied by urban expansion was analysed using the contribution index. Based on a model comparison, the geographically weighted regression (GWR) model was used to explore the spatio-temporal pattern and evolution path of significant driving factors. The results demonstrate that (1) the total area of cultivated land occupied by urban expansion from 2000 to 2018 was 470.528 km$^2$ and mainly concentrated in the main urban area of Chengdu City and its surrounding districts and counties. The K value continued to decline from 93.23 to 37.48, indicating that the contribution of cultivated land in urban expansion is decreasing. (2) The GWR model, with a better fitting effect, demonstrates that the significant factors that influence the cultivated land occupied by urban expansion in the study area gradually shift from population aggregation factors to food safety and the proportion of non-farming population. (3) The evolution type of the driving force is mainly dominated by continuous change, and there is significant spatial heterogeneity in the evolution path. The population density → proportion of non-farm population → grain yield → comprehensive and proportion of cultivated land → per capita fixed asset investment → comprehensive → proportion of non-farm population were typical pathways. (4) Realizing the coordination between urban system structural optimization and cultivated land protection policies in the Chengdu metropolitan area is an important way to guarantee regional food and ecological security and promote the green and high-quality development of the metropolitan area. This study can promote coordination between urban expansion and cultivated land protection in metropolitan areas and provide a reference basis for sustainable economic and social development.

**Keywords:** urban expansion; cultivated land occupation; driving force; evolution path; geographically weighted regression; Chengdu metropolitan area

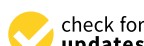



## 1. Introduction

Urban areas are those that have the strongest and most concentrated influences of human activities on the environment. Urban space expansion is not only irreversible and the most important form of land use but is also one of the most direct manifestations of urbanization [1,2]. As urbanization continues, rapid urban land expansion has begun to have a strong influence on surrounding land areas. As the primary land source for urban expansion, cultivated land is continuously occupied by urbanization. This may result in considerable losses of high-quality cultivated land, further intensifying the increasing supply–demand issues of cultivated land and influencing stable agricultural development

as well as food security [3,4]. According to available data, the cultivated land area will decrease by 1.8–2.4% due to the global urban expansion from 2000 to 2030, reducing global crop yields by about 3–4% [5]. China is a developing country and has the largest population in the world. Urbanization and cultivated land protection in China are of great concern for all countries. In the past three decades, China has experienced large-scale urban expansion, and the urbanization rate increased quickly from 26.4% in 1990 to 63.9% in 2020 [6,7]. A total of 71.16% of this urban expansion was attributed to the occupation of cultivated land. On the one hand, the large-scale occupation of cultivated land in China changed the landscape pattern in the region and intensified the scarcity of cultivated land resources. On the other hand, it degraded ecosystem service functions and had an important effect on the ecological environment and food security [8].

Urbanization is an inevitable trend of social and economic development. A large number of the rural population moved to cities, which promoted the rapid expansion of cities and needed more space to support urban development. Land is an essential resource for the survival and continued expansion of cities [9]. As a special natural resource, cultivated land has the characteristics of scarcity and the area is decreasing [10]. Understanding the dynamic changes in cultivated land resources from the perspective of rapid urbanization is crucial to assure future food security [11]. The occupation of cultivated land for urban expansion has become a focus of academic research. In terms of their research scale, existing studies have focused on the global scale [12,13], national scale [11,14], regional scale [15,16], basin scale [17], and so on. Moreover, an increasing number of multi-scale studies have also been reported. Nkeki et al. (2018) analysed land use transformation and urban growth characteristics in Benin metropolitan region, Nigeria, finding that the land use transition comprised urban and forest land occupying large-scale cultivated land [18]. Liu et al. (2019) quantized the cultivated land loss caused by China's urban expansion at the national scale, administrative level, population scale, and urban scale, ascertaining that China has suffered from considerable cultivated land losses during urban expansion since the 1970s [19]. Research methods for cultivated land occupation during urban expansion include the remote sensing interpretation method [20], night-time light data method [21], street-based metrics [22], regression model method [23], and scenario simulation [24,25]. In terms of the research content, scholars have focused on information extraction [26], spatiotemporal pattern evolutionary characteristics [27,28], driving forces and mechanisms [29,30], ecological environmental effects [31,32], economic and social effects [33–35], future scenario simulation [24], and farmers' livelihoods, satisfaction, and happiness [36,37] of cultivated land occupied by urban expansion. Erasu and Lika (2022) found that the urban expansion has led to a significant reduction in cultivated land, and more than 90% of urban expansion has been attributed to cultivated land in Akaki Kaliti sub-city from 1986 to 2019 [38]. Huang et al. (2019) studied the influences of China's urbanization on cultivated land area, proposing that it led to a higher cultivated land loss rate; land financing and urban sprawl caused by urbanization are major causes of cultivated land occupied by urban construction [39]. Huang et al. (2020) carried out a multi-scale analysis of global urban expansion from 1992 to 2016, as well as its influences on the net primary productivity (NPP) of cultivated land; they found that the total cultivated land losses caused by global urban expansion accounted for 45.9% of the total global urban expansion area and the relevant total NPP loss of cultivated land amounted to 58.71 TgC [10]. Rimal et al. (2019) used historical CA-Markov model to predict urban expansion for the years 2026 and 2036, and the results showed that the urban land will increase to 8.95% and 12.45%, respectively, while corresponding declines in cultivated land to 56.86% and 53.77% [40]. Among them, research on driving force and mechanism has always been a research hotspot of cultivated land occupied by urban expansion. Urban expansion occupying cultivated land is the result of the combined effect of economic, social, and political driving forces, and the driving mechanism is dynamic [41]. The regression model method is often used to discuss the driving factors of cultivated land occupied by urban expansion. Specifically, ordinary least squares (OLS) provide the method for modelling multiple linear relations, which provide

advantages over classical regression analysis [42]. Geographically weighted regression (GWR) is a local regression method that takes the spatial weight into account, and it can reveal the spatial heterogeneity of influencing factors and their explanatory ability [43]. Studying the spatio-temporal evolutionary characteristics of cultivated land occupied by urban expansion under severe land use conditions, in addition to disclosing the driving mechanisms scientifically, are crucial to relieve issues such as population stress, maintaining regional food and ecological security, and promoting the sustainable development of urban areas [22]. Nevertheless, existing studies on the driving mechanisms of cultivated land occupied by urban expansion are mainly static studies. They lack comprehensive discussions on the evolutionary paths of the driving mechanism. It is difficult to ascertain the dominant driving forces of occupying cultivated land for urban expansion in different periods, as well as their evolutionary trends. Hence, these studies have limited significance for guiding reasonable urban expansion and cultivated land protection.

Metropolitan areas are those that witness the most significant influences of urbanization [44]. The phenomenon of urban expansion occupying cultivated land in the metropolitan area is more prominent, and the contradiction between people and land is more serious. It is urgent to study the determining dominant driving forces of cultivated land loss and evolution paths during urbanization. Therefore, this study revealed the spatio-temporal variation laws of cultivated land occupied by urban expansion based on the current land use patterns in the Chengdu metropolitan area from 2000 to 2018. A regression model between the driving factors and cultivated land area occupied by urban expansion was constructed using the OLS regression and GWR model, and the fitting results of the regression models were compared. Moreover, a quantitative analysis regarding the evolutionary path and spatial heterogeneity of the significant driving forces of cultivated land occupied by urban expansion was carried out. It is not only helpful to promote coordination between urban expansion and cultivated land protection in the Chengdu metropolitan area, but also provides ideas to guarantee regional food and ecological security and realize sustainable economic and social development.

## 2. Materials and Methods

### 2.1. Study Area

Chengdu metropolitan area (Figure 1) is located in the southeast region of Sichuan Province (102°54′–105°01′ E, 29°24′–31°42′ N). It has four prefecture-level cities (Chengdu City, Deyang City, Meishan City, and Ziyang City) and 35 subordinated districts (cities and counties). It covers a total area of 33,114 km². Chengdu metropolitan area is an important junction between China's Yangtze River Economic Belt and China's Belt and Road Initiative and is an important economic centre in Southwest China. The terrain is generally high in the west and low in the east. The Longmen Mountains lie in the northwest and the Longquan Mountains lie in the centre, while the rest of the areas are mainly plains, accompanied by low hills and flat landforms. The region experiences a subtropical monsoon wet climate, typically characterised by warm and wet weather, along with frequent rainstorms in summer. Purple soil and paddy soil are dominant, which are high-quality soils. Many rivers, such as the Tuojiang River and Minjiang River, run through it, leading to a developed water system. Chengdu metropolitan area has the highest population density in Southwest China. By the end of 2019, the permanent population in the Chengdu metropolitan area was 24.19 million and the GDP per capita was 92,401.98 yuan, which is 65.67% higher than that in Sichuan Province. Rapid urbanization has led to the intensification of cultivated land being occupied by urban space expansion. The contradiction between urban development and cultivated land protection has become increasingly severe in recent years. Studying the spatial characteristics and driving forces of cultivated land occupation in the Chengdu metropolitan area during urbanization is important for food security and socio-economic development in Southwest China [45].

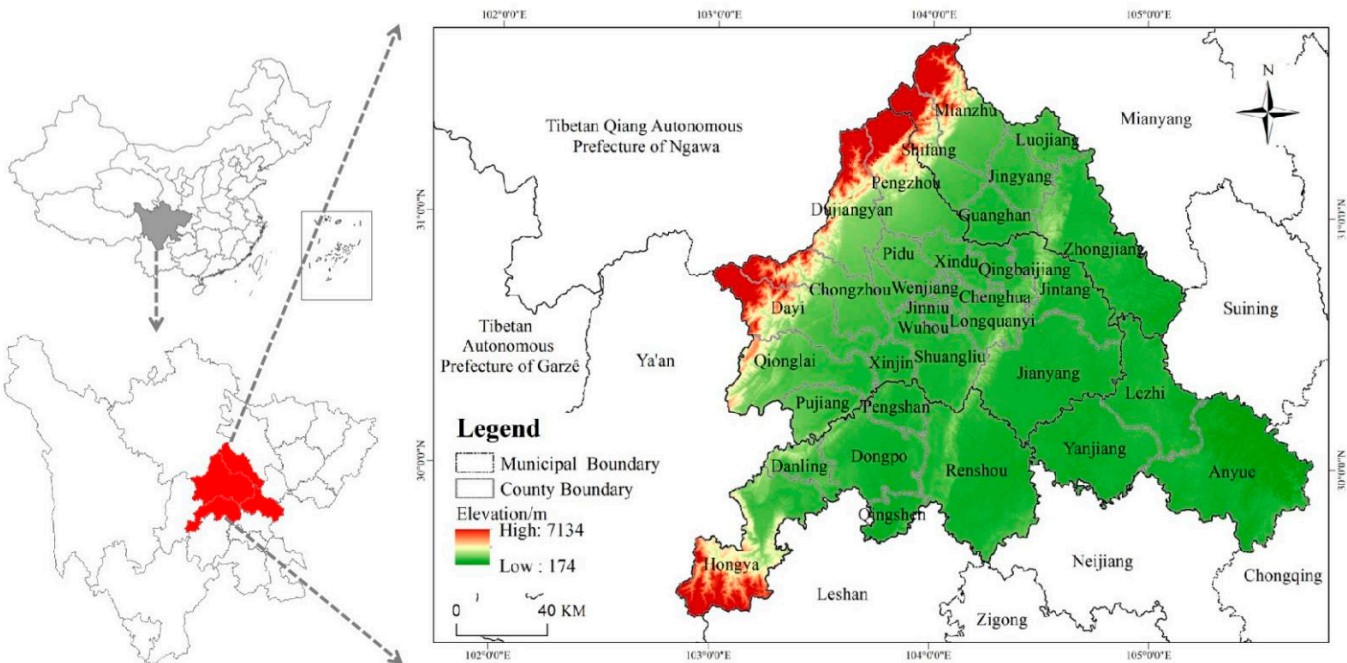

**Figure 1.** Geographical location of the study area.

### 2.2. Data Sources and Processing

Land use data and administrative region vector boundary data in the Chengdu metropolitan area from 2000 to 2018 were obtained from the Resources and Environment Science and Data Center, Chinese Academy of Sciences (https://www.resdc.cn/ (accessed on 15 December 2020)). The 30 m × 30 m digital elevation model (DEM) data was from the Geospatial Data Cloud platform (https://www.gscloud.cn/ (accessed on 20 December 2020)). The social and economic data, used to understand the driving forces, were from the Statistical Yearbook of Sichuan Province, China City Statistical Yearbook, and the China County Statistical Yearbook from 2001 to 2019. Based on data on cultivated land and construction land in the study area in 2000, 2005, 2010, 2015, and 2018, the dataset regarding cultivated land occupied by urban expansion in four periods was extracted. After the driving force dataset was collected, SPSS was applied for data normalization and the interpolation of missing data.

### 2.3. Methodology

#### 2.3.1. Technical Flowchart

First, based on the land use data and contribution index (*K*), the spatial and temporal evolution pattern of cultivated land occupied by urban expansion was analyzed, and the contribution of cultivated land in urban expansion was clarified. Second, from the dimensions of social elements, economic elements, traffic elements, and policy elements, the evaluation index system of cultivated land occupied by urban expansion was constructed. Third, the screened factors were input into the GWR model and OLS model to obtain the model with the best fitting effect. Last, based on the regression results of the optimal model, the driving force types and evolution paths of cultivated land occupied by urban expansion were analyzed. The technical framework is shown in Figure 2.

#### 2.3.2. Construction of the Driving Force Factor System

By using existing research results [46,47] and combining them with the availability and validity of data, a total of 11 indicators for traffic elements, policy elements, social elements, and economic elements were collected to establish the driving force system (Table 1).

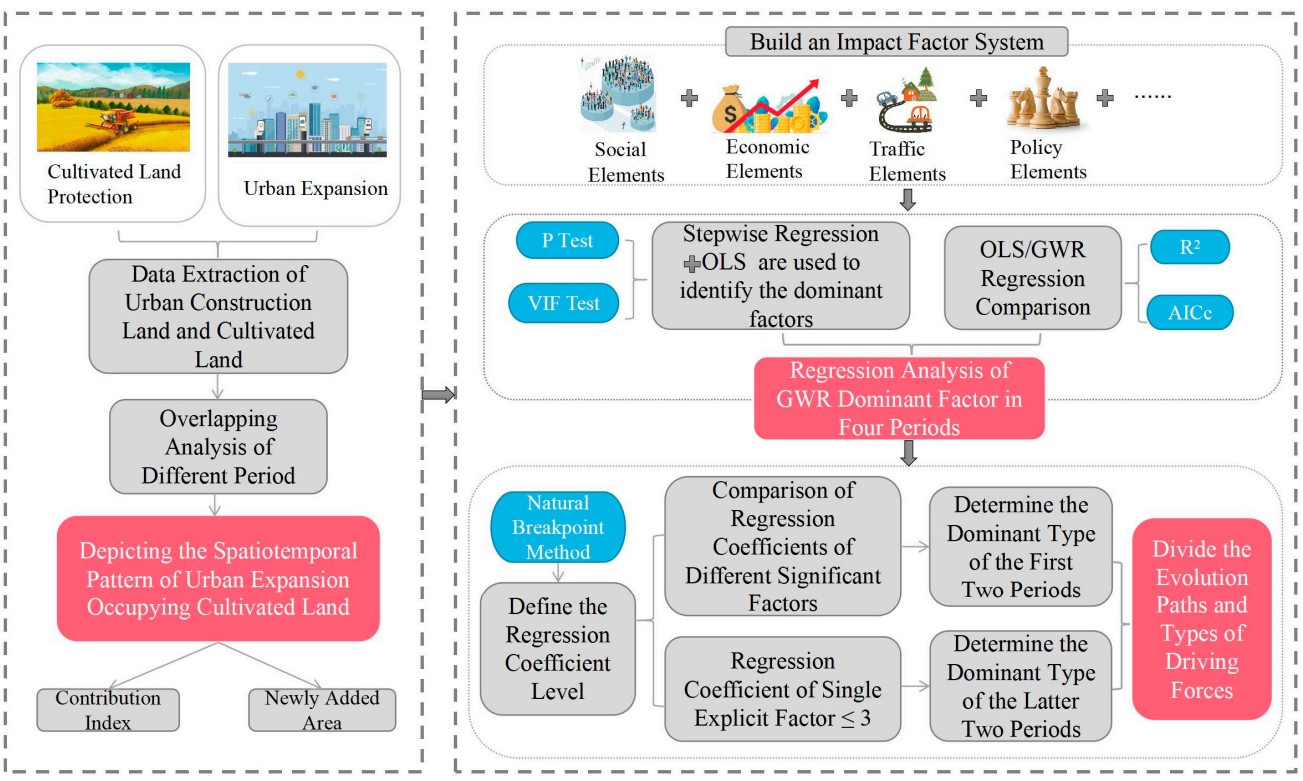

**Figure 2.** Technical flowchart.

**Table 1.** The driving force system of cultivated land occupied by urban expansion.

| Target Layer | Criterion Layer | Specific Indicators | Indicator Explanation |
|---|---|---|---|
| Driving force system of urban expansion occupying cultivated land | Social elements | Proportion of non-farm population | Non-farm population/Total population at the end of the year |
| | | Population density (Ten thousand people/km$^2$) | Permanent population at the end of the year/Land area |
| | Economic element | Grain yield (Tons) | – |
| | | Proportion of cultivated land | Cultivated land area/Total land area |
| | | Rural electricity consumption | – |
| | | Per capita GDP (Yuan/Person) | Regional GDP/Permanent population at the end of the year |
| | | Economic non-agriculturalization rate | Total output value of secondary and tertiary industries/Regional GDP |
| | | Total retail sales of per capita social consumer goods (Ten thousand yuan/km$^2$) | Total retail sales of consumer goods/Total land area |
| | Traffic elements | Road density (km/km$^2$) | Highway mileage/Total land area |
| | Policy elements | Per capita fiscal expenditure (Ten thousand yuan/ Ten thousand people) | General budget expenditure for local finances for the year/Regional total population at the end of the year |
| | | Total per capita fixed asset investment (Ten thousand yuan/km$^2$) | Social fixed assets investment/Total land area |

### 2.3.3. Measurement of Land Use Changes

The contribution index (*K*) [48] refers to the proportion of the occupied cultivated land in the newly added urban construction land area in different periods, which reflect the dependence of urban expansion on cultivated land occupation and the contribution of

cultivated land to urban expansion. So, *K* was used to evaluate the dynamic changes in urban expansion occupying cultivated land from 2000 to 2018.

$$K = \frac{C_a}{B_a} \times 100\% \tag{1}$$

where $B_a$ is the newly added urban construction land area in a given period, and $C_a$ is the occupied cultivated land area in the newly added urban construction land.

### 2.3.4. OLS Model

The OLS model is a typical global linear regression method. Its basic principle involves choosing a group of linear function parameters of explanatory variables according to the least-squares method to minimize the square sum of the distances from various points to the straight line [42].

$$y_i = \beta + \beta_0 x_0 + \beta_1 x_1 + \beta_2 x_2 \ldots + \beta_i x_i + \varepsilon_i \tag{2}$$

where $\beta$ is a constant, $y_i$ is the fitting value of cultivated land area occupied in different periods, and $\beta_0, \beta_1, \beta_2 \ldots \beta_i$ are the regression coefficients of different factors $x_1, x_2, x_3 \ldots x_i$.

### 2.3.5. GWR Model

Both OLS and GWR models are classical methods used to study driving factors, which have different abilities in terms of disclosing their performance. To select the optimal model to understand the spatio-temporal evolutionary characteristics of driving factors of cultivated land occupied by urban expansion in the study area, this study used the GWR model to discuss the evolutionary characteristics of driving forces. Moreover, the simulation results of the models were compared. The GWR model can be used to explore the spatial relationship between the explained variables and several independent variables [43]. In this study, the spatial heterogeneity of significant influencing factors of cultivated land occupied by urban expansion in 35 districts (cities and counties) of the Chengdu metropolitan area during different periods was explored using the GWR model.

$$y_i = \beta_0(u_i, v_i) + \sum_{j=1}^{k} \beta_j x_{ij} + \varepsilon_i \tag{3}$$

where $\beta_0(u_i, v_i)$ is the intercept of the function, $x_{ij}$ is the observed value of region $i$, $\beta_j$ is the influencing factor coefficient, and $\varepsilon_i$ is the error term.

We selected the Gauss function as the spatial weight function, and the calculation formula is as follows:

$$w_{ij} = \exp(-(d_{ij}/b)^2) \tag{4}$$

$w_{ij}$ is the observation weight at $i$, $d_{ij}$ is the distance between the regression point $i$ and the data point $j$, and the non-negative parameter $b$ is the bandwidth, which represents the functional relationship between the weight and the distance.

During the discrimination of the dominant factors, the regression coefficients of different significant factors were divided into seven influence levels using the natural breakpoint method. If two significant factors of a district (city and county) have the same influence levels, both are dominant factors. If one significant factor has a higher influence level, it is the dominant factor.

## 3. Results

### 3.1. Spatio-Temporal Pattern Analysis of Cultivated Land Occupied by Urban Expansion

3.1.1. Temporal Variation

The cultivated land occupied by urban expansion from 2000 to 2018 increased first and then decreased. Specifically, the cultivated land area occupied by urban expansion in the Chengdu metropolitan area increased continuously from 2000 to 2010, and then decreased from 2010 to 2018 (Table 2). The occupied cultivated land area from 2010 to 2018 decreased significantly in comparison to that in the previous period, but the

growth rate was negative from 2010 to 2015. The K value of cultivated land occupied by newly added urban construction land generally exhibited a decreasing trend from 2000 to 2018. This was because, with the rapid economic development in the study area, urban construction expanded quickly, resulting in losses in the labour population and the significant abandonment of cultivated land in rural areas. Meanwhile, the government of Sichuan Province advocated for the cultivated land balance policy during urban expansion to guarantee local food security. This decreased the embezzlement of cultivated land resources through urbanization.

**Table 2.** Statistics on urban expansion dynamics occupying cultivated land from 2000 to 2018.

| Period | Newly Added Urban Construction Land Area (km$^2$) | Occupied Cultivated Land Area in the Newly Added Urban Construction Land (km$^2$) | K |
|---|---|---|---|
| 2000–2005 | 184.550 | 172.057 | 93.23 |
| 2005–2010 | 233.204 | 206.916 | 88.73 |
| 2010–2015 | 26.828 | 20.989 | 78.24 |
| 2015–2018 | 188.256 | 70.566 | 37.48 |

### 3.1.2. Spatial Heterogeneity

To visualize the occupation of cultivated land areas in districts (cities and counties) more intuitively, the occupied cultivated land areas over four periods have been expressed in diagrams (Figure 3).

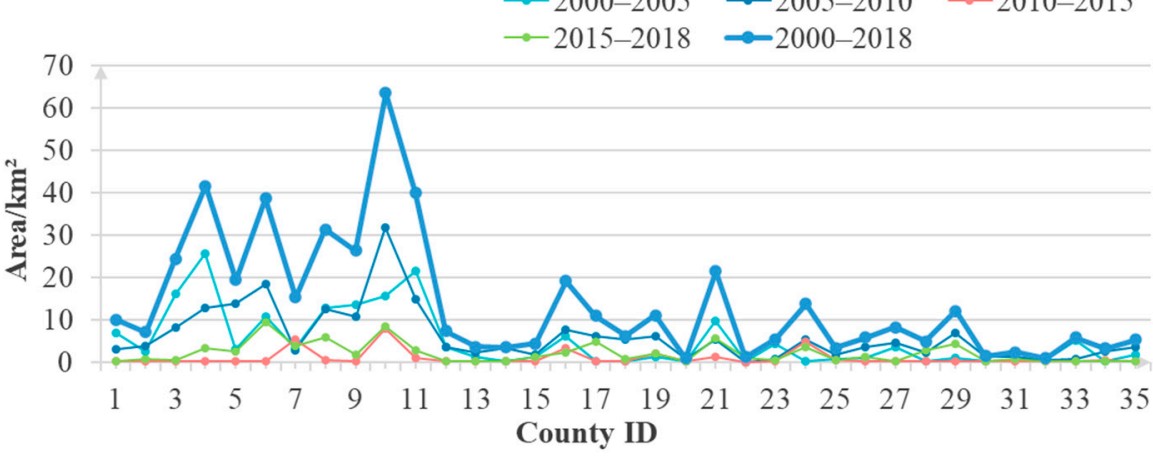

**Figure 3.** Variation of the cultivated land occupied by urban expansion in the study area from 2000 to 2018. Notes: 1–35 represents Jinjiang District, Qingyang District, Jinniu District, Wuhou District, Chenghua District, Longquanyi District, Qingbaijiang District, Xindu District, Wenjiang District, Shuangliu District, Pidu District, Jintang County, Dayi County, Pujiang County, Xinjin County, Dujiangyan City, Pengzhou City, Qionglai City, Congzhou City, Jianyang City, Jingyang City, Luojiang City, Zhongjiang County, Guanghan City, Shifang City, Mianzhu City, Dongpo District, Pengshan District, Renshou County, Hongya County, Danling County, Qingshen County, Yanjiang District, Anyue County, and Lezhi County, respectively.

Generally speaking, urban expansion occupying cultivated land in 35 districts (cities and counties) presented similar variation trends from 2000 to 2018. Eight main urban districts in Chengdu City and surrounding districts (cities and counties), including Chenghua District, Jinniu District, and Wuhou District, were the major areas of urban expansion occupying cultivated land. Among them, Shuangliu District was a typical example, and the occupied cultivated land area exceeded 120 km$^2$, followed by Wuhou District and Pidu District (>80 km$^2$). Jinjiang District, Dujiangyan City, Chongzhou City, Jingyang District,

Guanghan City, and Renshou County were secondary areas of urban expansion occupying cultivated land (>20 km$^2$). The cultivated land area occupied by urban expansion in Jianyang City, Luojiang City, and Qingshen County was the lowest. The cultivated land area occupied by urban expansion in the study area changed the most in two periods, 2000–2005 and 2005–2010, and was mainly concentrated in Wuhou District, Longquanyi District, Xindu District, Wenjiang District, and Shuangliu District. Specifically, the occupied cultivated land area increased year to year from 2005 to 2010, which was attributed to post-disaster construction. After 2010, the occupied cultivated land areas in districts (cities and counties) began to decrease. Variations in the occupied cultivated land area were similar during 2010–2015 and 2015–2018. Wenjiang District, Shaungliu District, Longquanyi District, Guanghan City, and Renshou County were the main occupation areas. After 2015, China's urbanization changed from external expansion to internal filling. In 2016, Chengdu metropolitan area became the core of the economic circle of Chengdu City and Chongqing, which facilitated regional urbanization. As a result, the occupied cultivated land area increased gradually.

### 3.2. Model Simulation Effect Evaluation and Screening of Significant Driving Forces

#### 3.2.1. Comparison of Model Fitting Effects

Collinearity among different indicators might lead to the inaccurate estimation of model parameters and poor precision in model fitting results. Therefore, a collinearity diagnosis is imperative before inputting the driving forces in Table 2 into the OLS and GWR models. According to this diagnosis, the VIF values of 11 driving factors were smaller than 7.5, and no collinear redundancy was found. Hence, the driving factors were input into the OLS and GWR models for regression analysis. The model results in different periods are listed in Table 3.

**Table 3.** Model results.

| Model Type | Period | $R^2$ | Adjust $R^2$ | AICc |
|---|---|---|---|---|
| OLS | 2000–2005 | 0.473 | 0.220 | 251.902 |
| | 2005–2010 | 0.503 | 0.266 | 247.410 |
| | 2010–2015 | 0.653 | 0.498 | 144.340 |
| | 2015–2018 | 0.358 | 0.255 | 187.228 |
| GWR | 2000–2005 | 0.636 | 0.467 | 250.581 |
| | 2005–2010 | 0.604 | 0.526 | 242.430 |
| | 2010–2015 | 0.653 | 0.487 | 139.365 |
| | 2015–2018 | 0.659 | 0.552 | 184.242 |

According to a comparison of the $R^2$ and the adjusted $R^2$ values, as well as the AICc values of the OLS and GWR models in four periods, the fitting degree of the GWR model was higher and 11 independent variables have a better explanatory ability for the increased urban expansion occupying cultivated land in all periods. Therefore, the fitting results of the GWR model were chosen to analyse the spatial heterogeneity of driving forces of urban expansion occupying cultivated lands; the optimal fixed bandwidth value was 2,214,450.498.

#### 3.2.2. Screening of Significant Driving Forces

In order to select the factors with strong significance and no multicollinearity to input into the GWR model, the significant driving force in the four periods were obtained by combining SPSS stepwise regression and OLS regression results (Table 4). All indicators passed the significance test of $p < 0.05$. To increase the analysis precision of the driving force, spatial heterogeneity was discussed using the regression coefficients of significant factors generated by the GWR model.

**Table 4.** Major driving forces in different periods.

| Period | Significant Factor | *p* |
|---|---|---|
| 2000–2005 | Population density | 0.030 |
| | Proportion of cultivated land | 0.003 |
| 2005–2010 | Proportion of non-farm population | 0.010 |
| | Total per capita fixed asset investment | 0.047 |
| 2010–2015 | Grain yield | 0.003 |
| 2015–2018 | Proportion of non-farm population | 0.026 |

### 3.3. Evolutionary Path Analysis of Driving Forces

#### 3.3.1. Dominant Factors in the Four Periods

From 2000 to 2005, the study area was mainly affected by the proportion of cultivated land, involving 17 districts (cities and counties), including Hongya County, Dujiangyan City, etc. (Figure 4a). This was mainly because these districts (cities and counties) were near mountains, and the proportion of cultivated land was slightly restricted by the geography. Hence, the large proportion of cultivated land inhibited the urban expansion more than the population density. However, the population density influences the urban expansion more in 16 districts (cities and counties), including Jinjiang District and Jinniu District. Both population density and the proportion of cultivated land influence the urban expansion in combination in Wuhou District and Shuangliu District.

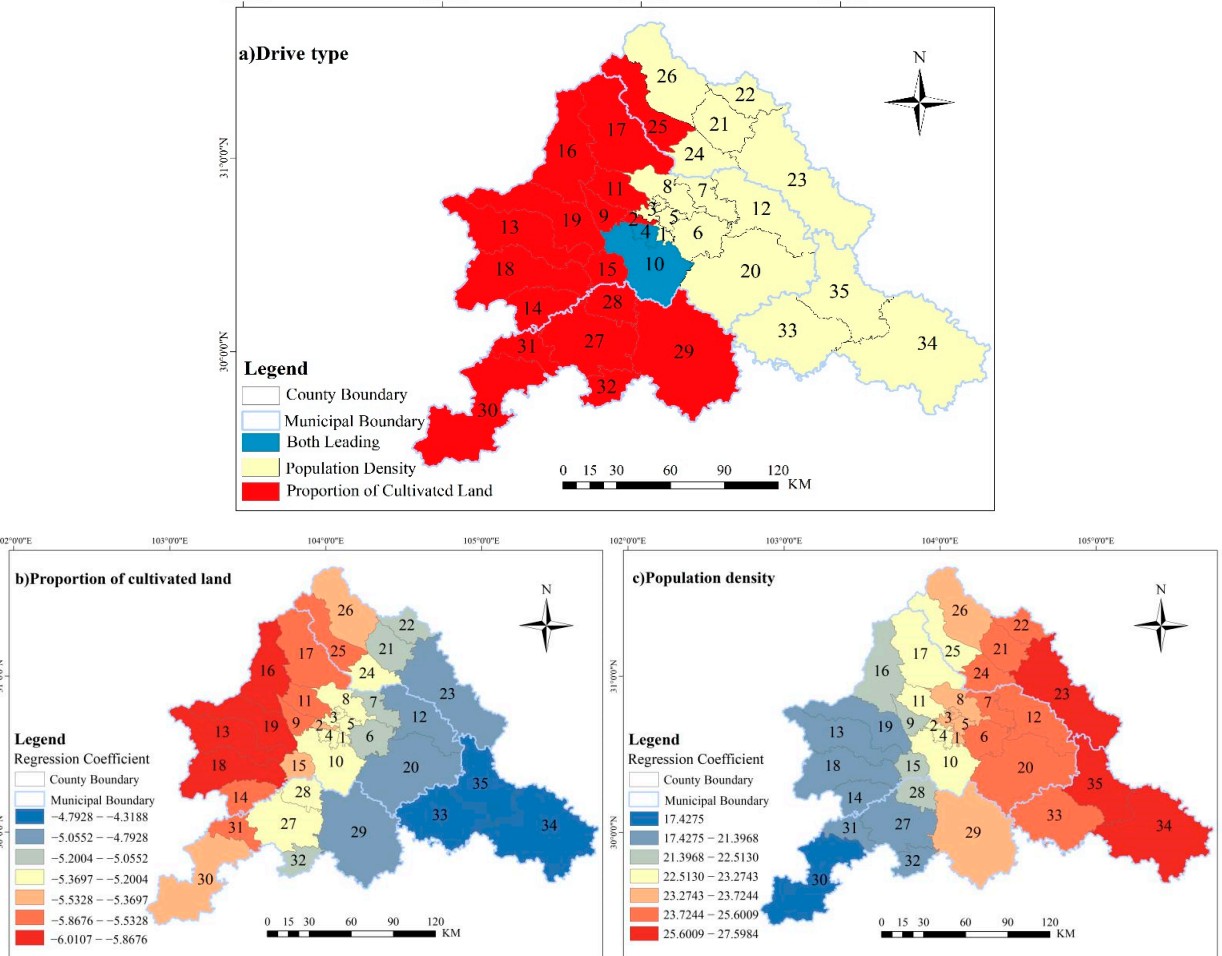

**Figure 4.** Dominant factors and driver types from 2000 to 2005: (**a**) Distribution pattern of dominant drive types; (**b**) Regression coefficient of the proportion of cultivated land; (**c**) Regression coefficient of population density.

Specifically, the proportion of cultivated land was negatively correlated with the occupation of cultivated land in the study area. The absolute value of the regression coefficient increased gradually from the southeast to the northwest, and the maximum value was concentrated in the western region, including Dujiangyan City, Congzhou City, Dayi County, and Qionglai City (Figure 4b). The regression coefficient of population density opposed the proportion of cultivated land (Figure 4c). A low population density was found in the western mountainous regions, which was high in the eastern plains. The regression coefficient increases from the west to the east, exhibiting positive correlations. Furthermore, large-scale complex terrain is present in Hongya County, which is disadvantageous for a concentrated population distribution, resulting in the low occupation of cultivated land. Since there is a large administrative scope in Zhongjiang County, Lezhi County, and Anyue County, the main urban areas of central Chengdu City are highly attractive for surrounding areas, and the population density in the middle and east regions is relatively high.

From 2005 to 2010, regions dominated by total per capita fixed asset investment included 12 districts (cities and counties), such as Hongya and Pujiang (Figure 5a). Regions driven by a proportion of non-farm population and per capita fixed asset investment included nine districts (cities and counties), such as Jianyang, Wenjiang, and Lezhi. The proportion of non-farm population influenced urban expansion occupying cultivated land more than the other indicators in six districts (cities and counties) of Deyang District as well as Dujiangyan, Mianzhu, and other regions (cities and counties) in northern Chengdu City. This is closely related to the accelerated population urbanization during post-disaster reconstruction. Specifically, the proportion of non-farm population and total per capita fixed asset investment were both positively related to urban expansion occupying cultivated land. In contrast, the regression coefficient of the non-farm population decreased from east to west (Figure 5b), while the regression coefficient of per capita fixed asset investment decreased from north to south (Figure 5c). The governments of Hongya County, Qingshen County, Renshou County, and Ziyang City increased local fixed asset investment continuously in this period. In Dujiangyan, Pengzhou, Shifang, Mianzhu, and Luojiang, the total fixed asset investments were slightly correlated with the urban expansion occupying cultivated land, and the investments were lower than those in other districts (cities and counties). Moreover, the fixed asset investment of these regions mainly depended on external aid reconstructions, but the proportion of the non-farm population increased more than in other regions. Therefore, the proportion of the non-farm population influenced the cultivation land occupation in these regions more than the other indicators.

From 2010 to 2015, grain yield was negatively correlated with urban expansion occupying cultivated land. The absolute value of the regression coefficient increased from south to north, which indicates that the inhibition effect of grain yield on urban expansion occupying cultivated land increased from south to north (Figure 6). The absolute value of the regression coefficient was high in Dujiangyan City, Pidu District, Xindu District, Jintang County, and Qingbaijiang District, indicating that the grain yield had a strong inhibition effect on urban expansion occupying cultivated land. However, this inhibition effect was the weakest in Hongya County and Qingshen County. In comparison to the trend during 2005–2010, the regression coefficient of the proportion of the non-farm population during 2015–2018 also increased from north to south gradually, exhibiting a positive correlation trend. This revealed that the proportion of the non-farm population made a greater contribution to urban expansion occupying cultivated land in Hongya County, Qingshen County, Renshou County, Yanjiang District, and Anyue County than other districts (cities and counties).

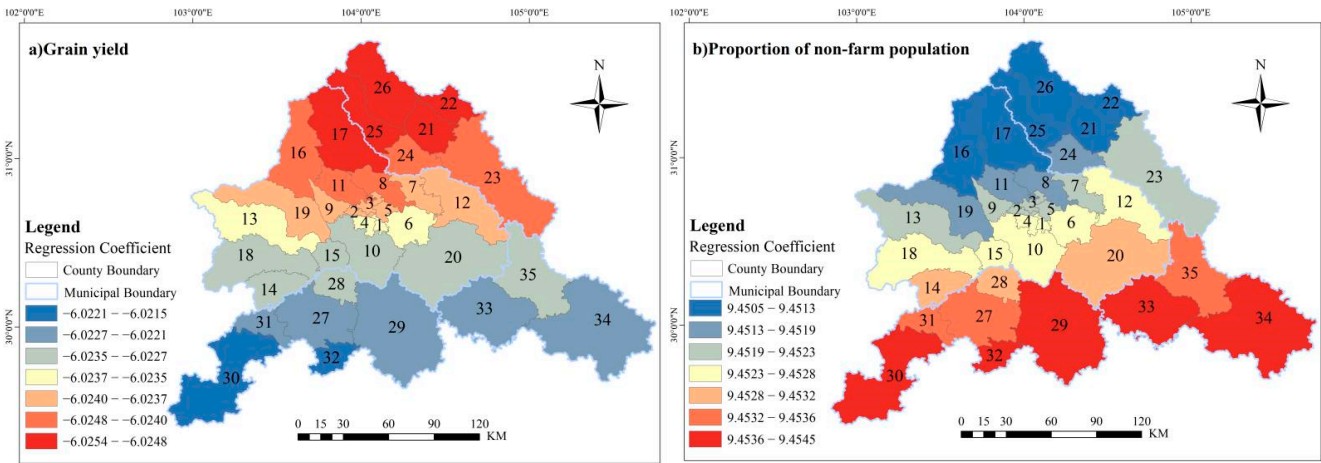

**Figure 5.** Dominant factors and driver types from 2005 to 2010: (**a**) Distribution pattern of dominant driver types; (**b**) Regression coefficient of the proportion of the non-farm population; (**c**) Regression coefficient of per capita fixed asset investment.

**Figure 6.** Dominant factors from 2010 to 2018: (**a**) Regression coefficients of grain yield (2010–2015); (**b**) Regression coefficient of the proportion of the non-farm population (2015–2018).

### 3.3.2. Evolutionary Type and Path Analysis

According to the influence degrees of significant factors in different districts (cities and counties) during 2010–2015 and 2015–2018, the regression coefficients of significant factors were divided into seven levels by the natural breakpoint method. Regression coefficients at level-3 or lower were classified as the comprehensive drivers. In combination with the changes in the major driving forces over four periods, the scenario with different significant factors in all four periods was classified as a continuous change type. The scenario when significant factors in the first two periods were the same was determined as the first-slow-then-urgent type. The scenario when the same significant factors fluctuated at an interval over four periods was determined as the first-quick-then-slow type. The final evolution types and evolution paths of districts (cities and counties) could then be obtained (Table 5).

**Table 5.** Driving force evolutions in the Chengdu metropolitan area.

| Evolution Type | Number | County ID | Evolution Path |
|---|---|---|---|
| Continuous change | 33 | 1, 6 | population density → both leading → grain yield → proportion of non-farm population |
| | | 33 | population density → total per capita fixed asset investment → comprehensive → proportion of non-farm population |
| | | 3, 7, 8, 21, 22, 23, 24, 26 | population density → proportion of non-farm population → grain yield → comprehensive |
| | | 5 | population density → both leading → grain yield → comprehensive |
| | | 20, 34, 35 | population density → both leading → comprehensive → proportion of non-farm population |
| | | 10 | both leading → total per capita fixed asset investment → comprehensive → proportion of non-farm population |
| | | 2, 11, 16, 17, 25 | proportion of cultivated land → proportion of non-farm population → grain yield → comprehensive |
| | | 9, 19 | proportion of cultivated land → both leading → grain yield → comprehensive |
| | | 14, 15, 18, 27, 28, 29, 30, 31 | proportion of cultivated land → total per capita fixed asset investment → comprehensive → proportion of non-farm population |
| | | 13 | proportion of cultivated land → total per capita fixed asset investment → grain yield → comprehensive |
| | | 32 | proportion of cultivated land → proportion of non-farm population → comprehensive → proportion of non-farm population |
| First-slow-then urgent | 1 | 4 | both leading → both leading → grain yield → proportion of non-farm population |
| First-quick-then slow | 1 | 12 | population density → proportion of non-farm population → grain yield → proportion of non-farm population |

It can be seen from Table 5 that the continuous change type was the dominant evolution type in the study area, while Jinniu District and Jintang County were the first-slow-then-urgent type and the first-quick-then-slow type, respectively. Specifically, the dominant driving force in Deyang City and Chengdu City (e.g., Dujiangyan City) changed from population density and proportion of the non-farm population to the comprehensive driving type. During 2000–2005, the proportion of cultivated land was the dominant driving force in Meishan City. During 2005–2010, the proportion of the non-farm population was the dominant driving force in Qingsheng County, while per capita fixed asset investment was the dominant driving force in other regions. In the last two periods, the comprehensive evolution type changed to a dominant effect of the proportion of the non-farm population. In Chengdu City (cities and counties), the dominant driving force changed from population density to the proportion of the non-farm population. After 2010, with the strengthening focus on ecological security and cultivated land protection, grain yield changes in the middle and northern regions of Chengdu City further constrained the trend of urban expansion occupying cultivated land. During 2015–2018, the non-farm population in the western and

northern regions decreased due to the orientation of the west control strategy of the study area and its influences on the occupation of cultivated land decreased. The driving force became the comprehensive type. In all four periods, urban expansion occupying cultivated land in Yanjiang District of Ziyang City was mainly driven by economic elements such as population density and the proportion of the non-farm population. During 2000–2005 and 2005–2010, the driving force in Lezhi County and Anyue County changed from population density to both population density and the proportion of the non-farm population. In the follow-up urban expansion occupying cultivated land, the inhibition rate of grain yield weakened and the comprehensive driving force was changed to the positive influence of the proportion of the non-farm population.

## 4. Discussion

Cultivated land protection has become a major factor influencing sustainable social and economic development [49,50]. Some countries, such as China, the United States, Japan, etc., have implemented strict cultivated land protection policies to control the rapid conversion of cultivated land into urban construction land. In fact, there have been discussions and disputes about the cultivated land occupied by urban expansion. Some scholars believe that the cultivated land occupied by urban expansion is the inevitable result of urbanization, which is helpful to avoid the possible loss of environment and agricultural ecosystem, and the decline of agricultural production can be compensated by using modern technologies [51,52]. In some scholars' view, the occupation of cultivated land will cause the loss of agricultural ecosystems, affect agricultural production, and threaten food security and farmers' livelihood strategies [53]. Thus, it should be limited urgently. Some scholars have realized that although urbanization is an inevitable trend of social and economic development, it is necessary to protect cultivated land in the process of urban expansion [28,54]. However, large-scale cultivated land loss is still obvious under the effect of urban expansion. It restricts the regional food and ecological security, as well as sustainable development. Therefore, it is imperative to coordinate the relationship between urban expansion and cultivated land protection during urbanization.

Metropolitan areas are the core of the new era of urbanization and economic development, and they are irreplaceable in promoting global urbanization. Chengdu metropolitan area is an important economic centre in Southwest China. It is not only an important component of China's Yangtze River Economic Belt, but also a key national strategic development zone in China and has considerable development potential. With rapid progress in the urbanization of the Chengdu metropolitan area, the phenomenon of urban expansion occupying cultivated land is still prominent. Analysing the driving factors and evolution characteristics of the urban expansion occupying cultivated land is conducive to determine the dominant driving forces of cultivated land loss and evolution paths during urbanization. It can promote coordination between urban expansion and cultivated land protection and guide the metropolitan area toward reasonable new urbanization directions.

Changes in urban construction land and cultivated land are substantially a spatial response process of human production and life. The occupation of cultivated land during the urbanization of the Chengdu metropolitan area mainly expanded from central cities to surrounding districts (cities and counties). The occupation area of cultivated land and the K value decreased gradually, indicating that the contribution of cultivated land in urban expansion was decreasing. This is consistent with the work of Liu et al. [19]. However, Erasu and Lika, in their study of Ethiopia, found that the area of cultivated land occupied by urban expansion continued to increase, and the contribution of cropland to urban expansion also showed an increasing trend [38]. The reason why their results are inconsistent with this study may be that the Chinese government has implemented strict cultivated land protection and balance policies, and the cultivated land area occupied by urban expansion has been effectively controlled. With rapid social and economic development, population returns and policy factors became dominant driving forces of urban expansion occupying cultivated land in the metropolitan area. This was consistent with previous research results

of Zhong et al. [46], Guan et al. [47], and Dadashpoor et al. [55]. In view of the evolution types and evolution paths of driving forces in terms of urban expansion occupying cultivated land, Chengdu metropolitan area presented diversified spatial development features. Improving the capacity of urban planning is essential to reduce the negative effect of the urban expansion and ensure the food security [56]. To assure lasting and stable high-quality economic development, the urban system structure in the Chengdu metropolitan area shall be optimized to adapt to the cultivated land protection policy. On the one hand, attention must be paid to local spatial concentrations of different urban levels in metropolitan areas. On the other hand, coordination among urban spaces, agricultural spaces, and ecological spaces must be improved in the entire metropolitan area. Firstly, the core driving function of Chengdu City will occur fully and the urban integration with surrounding cities such as Deyang City, Ziyang City, and Meishan City will be accelerated. It is suggested to create a modernized metropolitan area that drives Sichuan, radiates to Southwest China, and has international influences. Secondly, urbanization boundaries should be determined reasonably under the premise of ecological protection and food security. This can help better coordinate territorial space planning in the region, realize the dynamic balance of cultivated land with surrounding areas, realize the intensive management of cultivated land gradually, and improve cultivated land benefits. Finally, the cooperation of cities must be promoted in metropolitan areas, strengthening the talent policy constructions and fixed asset investment in urban areas. This can help make full use of regional characteristics or dominant conditions to develop a local economy and increase the utilization of cultivated land occupied by urbanization. Moreover, it can develop the tourism industry by combining the urban–rural transition zone and increase cultivated land planting and other cultural travel experience modes to extend the length of the economic industrial chain.

However, this study has certain limitations. Firstly, this paper only studied the spatial heterogeneity of driving forces of cultivated land occupied by urban expansion, without considering the uncertainty caused by the temporal heterogeneity simultaneously. In the future, a GTWR model which can take into account the temporal and spatial heterogeneity of driving factors will be used to analyze the driving forces of cultivated land occupied by urban expansion. Moreover, only differing significant factors in different periods were chosen to analyse spatial heterogeneity. In the future, common significant factors in different periods should also be considered, and spatial evolution of the same factor at different time points should be analysed. Lastly, multiple industrial modes of urban expansion occupying cultivated land in the urban–rural transition zone in metropolitan areas, as well as the method to relate it to rural economic development, should be discussed from the perspective of rural revitalization. This can increase the utilization efficiency of occupied cultivated land and promote the relationship between urban expansion and ecological protection.

## 5. Conclusions

Based on data regarding cultivated land area occupied by newly added urban construction land in Chengdu metropolitan area from 2000 to 2018, the contribution index and spatial heterogeneity were analysed. The GWR model, which has a better model fitting effect, was used to analyse the evolution types and evolution paths of driving forces. Some major conclusions could be drawn:

(1) During 2000–2018, the cultivated land area occupied by urban expansion increased first and then decreased, and the contribution of cultivated land to urban expansion decreased continuously. In terms of the spatial distribution, cultivated land occupied by urban expansion in the study area was mainly concentrated in Chengdu urban areas (cities and counties). The centre moved from the main urban areas to surrounding districts (cities and counties). During 2005–2010, urban expansion occupying cultivated land mainly occurred in the eastern and southern regions (cities and counties), such as Shuangliu District, Wenjiang, Longquanyi District, and Renshou County.

(2) The GWR model is more appropriate to study the spatio-temporal heterogeneity of driving forces than the OLS model. In comparison to the OLS model, the AICc and adjusted R$^2$ of the GWR model were higher. In different periods, six factors—including the population density and proportion of cultivated land—influenced the urban expansion occupying cultivated land more significantly.

(3) In different periods, there is obvious spatial heterogeneity in the dominant driving forces for urban expansion occupying cultivated land. It can be divided into different evolution types and paths. During 2000–2005, the proportion of cultivated land was the dominant driving force and the negative influences strengthened from east to west. During 2005–2010, the total per capita fixed asset investment was the dominant driving force and the positive influences strengthened from north to south. During 2010–2015, grain yield was negatively related to the proportion of the occupied cultivated land. During 2015–2018, the positive influences of the proportion of the non-farm population also gradually increased from north to south. In the study area, the continuous change type generally became the dominant evolution type, while Wuhou District and Jintang County were the first-slow-then-urgent type and first-quick-then-slow type, respectively. With respect to evolution path, population density → proportion of non-farm population → grain yield → comprehensive and proportion of cultivated land → per capita fixed asset investment → comprehensive → proportion of non-farm population were typical pathways.

**Author Contributions:** Conceptualization, methodology, B.M. and P.H.; software, B.M., X.W. and Z.Z.; investigation, B.M. and X.W.; formal analysis, B.M., X.W. and Z.Z.; data curation, B.M. and P.H.; visualization, B.M., X.W. and Z.Z.; writing—original draft preparation, B.M., X.W. and Z.Z.; writing—review and editing, B.M., P.H. and X.W.; supervision, P.H.; funding acquisition, B.M. All authors have read and agreed to the published version of the manuscript.

**Funding:** This research was funded by supported by the high-level talent "QiHang" program of Yibin University (No. 2021QH037).

**Institutional Review Board Statement:** Not applicable.

**Informed Consent Statement:** Not applicable.

**Data Availability Statement:** Not applicable.

**Conflicts of Interest:** The authors declare no conflict of interest.

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
