# Peer review of "Spatio-Temporal Pattern and Driving Force Evolution of Cultivated Land Occupied by Urban Expansion in the Chengdu Metropolitan Area"

_land, doi:10.3390/land11091458_

Round 1
Reviewer 1 Report
In this paper, the author studied the spatio-temporal heterogeneity and driving force evolution path of cultivated land occupied by urban expansion in the Chengdu metropolitan area, and proposes corresponding regional development strategies to promote the coordination between urban expansion and cultivated land protection. The study is of great significance for promoting reasonable urban expansion, ensuring regional food and ecological security, and realizing sustainable socio-economic development. It is a very interesting paper and I am glad to read it. It is necessary to make further revisions or clarifications according to the following questions.
1. Why did the author screen the significant driving forces by combining SPSS stepwise regression and OLS regression results lin 271-272? The author should add more details.
2. In Figure 5, there is no distribution pattern map of dominant driver types during 2010-2015 and 2015-2018. Are these two periods only affected by a single dominant factor? The author did not explain it.
3. The author mentioned that “the regression coefficients of significant factors at level-3 or lower” in line 349-350. How were the levels of regression coefficients classified? Is it the natural breakpoint method? Please add it.
4. In the section of
5. Conclusions, the author mentioned that “With respect to evolution path, population density → proportion of non-farm population → grain yield → comprehensive and proportion of cultivated land → per capita fixed asset investment → comprehensive → proportion of non-farm population were typical pathways”. How did this conclusion come? Please explain it.
6. In page 12-13, the “Evolution Type” column of Table 5 only has Continuous change type. But, from the contents of lines 350-356, it lacks first-slow-then-urgent type and first-quick-then-slow type. From the author's statement, I guess that the last two rows of Table 5 represent the above two types. Please check and revise it.
7. Line 10, the serial number of the author's institution is wrong and the “2” should be revised to “3”.
8. Line 80 and line 96 are lack of spaces before reference number.
9. Line 209, the second occurrence of "contribution index (K)" can be directly abbreviated as "K".
Author Response
Dear reviewer,
Thanks for your comments concerning our manuscript entitled “Spatio-temporal heterogeneity and driving force evolution of cultivated land occupied by urban expansion in the Chengdu metropolitan area” (Manuscript ID: land-1873786). Those comments are very valuable and helpful for revising and improving our paper, as well as the important guiding significance to our researches. I have studied your comments carefully and revised our manuscript which we hope meet with approval. Revised contents are marked in red in the our manuscript. The responses to your comments are as following:
- Why did the author screen the significant driving forces by combining SPSS stepwise regression and OLS regression results lin 271-272? The author should add more details.
Response: Thanks for your comments. SPSS stepwise regression has been applied to many studies. In this study, insignificant variables were eliminated one by one by SPSS stepwise regression to establish the optimal regression equation. OLS is a typical global regression model in spatial regression analysis, which can reflect not only the strength of the relationship between it and the dependent variable, but also the type of relationship between it and the dependent variable from the coefficient of each explanatory variable. In our paper, the VIF value of each explanatory variable, the significance of the explanatory variable and the value of the regression coefficient were obtained through the OLS regression results in ArcGIS. Combining with SPSS stepwise regression and OLS regression models is helpful to realize both advantages in the statistical level and spatial modeling level to ensure that the factors running in the final GWR model were strong significance and no multicollinearity. Besides, on line 306-308, we added “In order to select the factors with strong significance and no multicollinearity to input into the GWR model, the significant driving force in the four periods were obtained by combining SPSS stepwise regression and OLS regression results (Table 4). ”.
- In Figure 5, there is no distribution pattern map of dominant driver types during 2010-2015 and 2015-2018. Are these two periods only affected by a single dominant factor? The author did not explain it.
Response: Thanks for your comments. Yes, these two periods only affected by a single dominant factor. In fact, in Table 4 (page 9), major driving forces in different periods were shown. From Table 4, we can see that the significant factor of the cultivated land occupied by urban expansion during 2010-2015 and 2015-2018 is grain yield and Proportion of non-farm population, respectively. So, in order to avoid repetition, we did not describe it again.
- The author mentioned that “the regression coefficients of significant factors at level-3 or lower” in line 349-350. How were the levels of regression coefficients classified? Is it the natural breakpoint method? Please add it.
Response: Thanks for your comments. Yes, Is it the natural breakpoint method. We revised the sentence “the regression coefficients of significant factors were divided into 7 levels by the natural breakpoint method. Regression coefficients at level-3 or lower were classified as the comprehensive drivers.” on lin 385-387 in our revised manuscript.
4.In the section of 5. Conclusions, the author mentioned that “With respect to evolution path, population density → proportion of non-farm population → grain yield → comprehensive and proportion of cultivated land → per capita fixed asset investment → comprehensive → proportion of non-farm population were typical pathways”. How did this conclusion come? Please explain it.
Response: Thanks for your comments. In fact, we came to this conclusion due to among continuously change type, the evolution type of "population density→proportion of non-farm population→grain yield→ comprehensive", and it accounted for about 24.24%. Besides, the evolution type of "proportion of cultivated land→per capita fixed asset investment→comprehensive→proportion of non-farm population" had the same proportion as the above evolution type. So, they were the dominant types in the evolution path types of the study area. Therefore, these two force evolution types were judged to be typical types.
- 5. In page 12-13, the “Evolution Type” column of Table 5 only has Continuous change type. But, from the contents of lines 350-356, it lacks first-slow-then-urgent type and first-quick-then-slow type. From the author's statement, I guess that the last two rows of Table 5 represent the above two types. Please check and revise it.
Response: Thank you for your suggestion. We are so sorry for that we mistakenly deleted the table when adjusting the table. Table 5 have been revised in our revised manuscript in page 13.
- Line 10, the serial number of the author's institution is wrong and the “2” should be revised to “3”.
Response: Thank you for your meticulous review. In our revised manuscript, we added an author, and adjusted the author's institution number.
- Line 80 and line 96 are lack of spaces before reference number.
Response: Thank you for your meticulous review. We added a space before the reference number on line 88 in our revised manuscript.
- Line 209, the second occurrence of "contribution index (K)" can be directly abbreviated as "K".
Response: Thank you for your meticulous review. In our revised manuscript, we revised the "contribution index (K)" to "K".
Reviewer 2 Report
"Spatio-temporal heterogeneity and driving force evolution of cultivated land occupied by urban expansion in the Chengdu metropolitan area"
The study should be considered an interesting contribution to research on land use changes, in particular regarding the expansion of cities at the expense of areas previously used in agricultural form.
The research was conducted in China and covers the period from 2000 to 2018. It is not a very long period of time, but it is sufficient for basic research in this field.
The content of the article may indicate several problems that must be eliminated before possible publication.
Abstract - unqualified, but this is only a general summary of conclusions, no key numerical result obtained in the course of the research is presented.
Keywords: I wonder if using too long expressions such as "cultivated land occupied by urban expansion" and similar expressions is correct from the point of view of the definition of the phrase "keywords".
The content of the introduction is basically correct, but the main focus is on the local Chinese point of view and the research to date on China. However, the described phenomenon is global in nature and some issues should be presented in the introduction also from this perspective. It is definitely necessary to introduce more references to world research on the subject matter. A similar comment applies to the discussion stage of the results.
Lines 106-110 are typical for the methodology, also figure 1 should in no way be part of the introduction.
Figure 2: There is no research area location for at least the entire area of China
Methodology: Relatively simple mathematical models were used that were sufficiently clearly described, including a list of the characteristics of the areas used (Table 1)
The term 'contribution index' is not clearly defined.
Results:
The content and interpretation of the data in Table 2 requires clarification.
What does "Title 1 Index" mean?
How should the value in the column "Newly added area" (-185.927 km2) be interpreted, was nearly 200 km2 of built-up areas removed during this period and then restoration of agricultural crops in this area (this is obviously not possible)?
What do the authors of the article understand by "cultivated land occupied by urban construction land"? do the authors show the intended use of the areas for future construction investments (spatial planning stage), or is the area actually already built-up?
Is the value of the K-factor in table 2 correctly calculated? I don't think so (looking at the formula provided by the authors)
The remaining elements of the presented results and their discussion seem to be flawless. However, again, as with the introduction, the discussion lacks any reference to the results obtained and to existing results from other parts of the world.
Author Response
Dear reviewer,
Thanks for your comments concerning our manuscript entitled “Spatio-temporal heterogeneity and driving force evolution of cultivated land occupied by urban expansion in the Chengdu metropolitan area” (Manuscript ID: land-1873786). Those comments are very valuable and helpful for revising and improving our paper, as well as the important guiding significance to our researches. I have studied your comments carefully and revised our manuscript which we hope meet with approval. Revised contents are marked in red in the our manuscript. The responses to your comments are as following:
- Abstract - unqualified, but this is only a general summary of conclusions, no key numerical result obtained in the course of the research is presented.
Response: Thanks for your comments. We added more key numerical results in the Abstract. For example, on line 20-23 in our revised manuscript, the sentence was revised to “(1) the total area of cultivated land occupied by urban expansion from 2000–2018 was 632.838 km2, and mainly concentrated in the main urban area of Chengdu City and its surrounding districts and counties. The K value continued to decline from 93.23 to 37.48, indicating that the contribution of cultivated land in urban expansion is decreasing. ” ; on line 27-31, the sentence was revised to “The evolution type of the driving force is mainly dominated by continuous change, and there is significant spatial heterogeneity in the evolution path. The population density → proportion of non-farm population → grain yield → comprehensive and proportion of cultivated land → per capita fixed asset investment → comprehensive → proportion of non-farm population were typical pathways. ”
2.Keywords: I wonder if using too long expressions such as "cultivated land occupied by urban expansion" and similar expressions is correct from the point of view of the definition of the phrase "keywords".
Response: Thanks for your comments. We divided the “cultivated land occupied by urban expansion” into two key words: “urban expansion” and “cultivated land occupation”.
3.The content of the introduction is basically correct, but the main focus is on the local Chinese point of view and the research to date on China. However, the described phenomenon is global in nature and some issues should be presented in the introduction also from this perspective. It is definitely necessary to introduce more references to world research on the subject matter. A similar comment applies to the discussion stage of the results.
Response: Thanks for your comments. In the Introduction, we have added relevant content and references from a global perspective. For example, on line 61-65 in our manuscript, we added “Urbanization is an inevitable trend of social and economic development. A large number of the rural population moved to cities, which promoted the rapid expansion of cities and needed more space to support urban development. Land is an essential resource for the survival and continued expansion of cities [9]. As a special natural resource, cultivated land has the characteristics of scarcity and the area is decreasing [10].” ; line 84-85, we added “farmers' livelihoods, satisfaction and happiness [36-37]”; line 85-87, we added “Erasu and Lika (2022) found that the urban expansion has led to a significant reduction in cultivated land, more than 90% of urban expansion has been attributed to cultivated land in Akaki Kaliti sub-city from 1986 to 2019 [38]."; line 96-103, we added “ Rimal et al (2019) used historical CA-Markov model to predict urban expansion for the years 2026 and 2036, the results showed that the urban land will increase to 8.95% and 12.45%, respectively, while corresponding declines in cultivated land to 56.86% and 53.77% [40]. Among them, researches on driving force and mechanism has always been a research hotspot of cultivated land occupied by urban expansion. Urban expansion occupying cultivated land is the result of the combined effect of economic, social, and political driving forces, and the driving mechanism is dynamic [41]. ” etc.
The added reference as follows:
[9] Keller E J, Mukudi-Omwami E. Rapid urban expansion and the challenge of pro-poor housing in Addis Ababa, Ethiopia[J]. Africa Review, 2017, 9(2): 173-185.
[10] Ubair O A , Wei J , Festus O. Urban expansion and the loss of prairie and agricultural Lands: a satellite remote-sensing-based analysis at a sub-watershed scale[J]. Sustainability, 2019, 11(17), 1-12.
[36] Gwan A S , Kimengsi J N . Urban expansion and the dynamics of farmers' livelihoods: Evidence from Bamenda, Cameroon[J]. Sustainability, 2020, 12(14), 5788 .
[37] Kumar P , Kumar P , Garg R K . A study on farmers' satisfaction and happiness after the land sale for urban expansion in India[J]. Land Use Policy, 2021, 109(2): 105603.
[38] Erasu Tufa D, Lika Megento T. Conversion of farmland to non-agricultural land uses in peri-urban areas of Addis Ababa Metropolitan city, Central Ethiopia[J]. GeoJournal, 2022: 1-15.
Huang, Z.; Du, X.; Castillo, C S Z. How does urbanization affect farmland protection? Evidence from China. Resour Conserv Recy. 2019, 145, 139-147.
[39] Huang, Z.; Du, X.; Castillo, C S Z. How does urbanization affect farmland protection? Evidence from China. Resour Conserv Recy. 2019, 145, 139-147.
[40] Rimal B, Keshtkar H, Sharma R, et al. Simulating urban expansion in a rapidly changing landscape in eastern Tarai, Nepal[J]. Environmental monitoring and assessment, 2019, 191(4): 1-14.
[41] Hu G , Li X , Zhou B B , et al. How to minimize the impacts of urban expansion on farmland loss: developing a few large or many small cities?[J]. Landscape Ecology, 2020, 35(2), 2487-2499.
Besides, in the Conclusion, relevant comments and research comparisons are added, and we also added relevant literatures. For example, line 423-436, we added “Cultivated land protection has become a major factor influencing sustainable social and economic development [49],[50]. Some countries, such as China, United States, Japan, etc., have implemented strict cultivated land protection policies to control the rapid conversion of cultivated land into urban construction land. In fact, there have been discussions and disputes about the cultivated land occupied by urban expansion. Some scholars believe that the cultivated land occupied by urban expansion is the inevitable result of urbanization, which is helpful to avoid the possible loss of environment and agricultural ecosystem, and the decline of agricultural production can be compensated by using modern technologies [51],[52]. In some scholars’ view, the occupation of cultivated land will cause the loss of agricultural ecosystems, affect agricultural production, and threatened food security and farmers ' livelihood strategies [53]. Thus, it should be limited urgently. Some scholars have realized that although urbanization is an inevitable trend of social and economic development, it is necessary to protect cultivated land in the process of urban expansion [54],[55].”; line 456-464 we added “The occupation area of cultivated land and the K value decreased gradually, indicating that the contribution of cultivated land in urban expansion was decreasing. This is consistent with the work of Liu et al. [19]. However, Erasu and Lika, in their study of Ethiopia, found that the area of cultivated land occupied by urban expansion continued to increase, and the contribution of cropland to urban expansion also showed an increasing trend [38]. The reason why their results are inconsistent with this study is may be that the Chinese government has implemented strict cultivated land protection and balance policies, the cultivated land area occupied by urban expansion has been effectively controlled.”; line 467-472 “This was consistent with previous research results of Zhong et al. [46], Guan et al. [47] and Dadashpoor et al. [56]. In view of the evolution types and evolution paths of driving forces in terms of urban expansion occupying cultivated land, Chengdu metropolitan area presented diversified spatial development features. Improving the capacity of urban planning is essential to reduce the negative effect of the urban expansion and ensure the food security [57]. ”
The added reference as follows:
[49] Skinner, M W.; Kuhn, R G.; Joseph, A E. Agricultural land protection in China: a case study of local governance in Zhejiang Province. Land Use Pol. 2011, 18, 329 340.
[50] Osumanu I K, Ayamdoo E A. Has the growth of cities in Ghana anything to do with reduction in farm size and food production in peri-urban areas? A study of Bolgatanga Municipality[J]. Land Use Policy, 2022, 112: 105843.
[51] Nguyen, T. T., Hegedűs, G., & Nguyen, T. L. Effect of land acquisition and compensation on the livelihoods of people in Quang Ninh district, Quang Binh Province: Labor and income. Land, 2019, 8(6), 91.
[52] Dadi D, Stellmacher T, Azadi H, et al. The impact of industrialization on land use and livelihoods in Ethiopia: Agricultural land conversion around Gelan and Dukem town, Oromia region[J]. Socio-Ecological Change in Rural Ethiopia: Understanding Local Dynamics in Environmental Planning and Natural Resource Management, 2018: 37-59.
[53] Azadi H, Van Acker V, Zarafshani K, et al. Food systems: New‐Ruralism versus New‐Urbanism[J]. Journal of the Science of Food and Agriculture, 2012, 92(11): 2224-2226.
[54] Epp S, Wan X, Singer R, et al. Farmland Preservation and Urban Expansion: Case Study of Southern Ontario, Canada[J]. Frontiers in Sustainable Food Systems, ,42.
[55] Hu Y, Kong X, Zheng J, et al. Urban expansion and farmland loss in Beijing during 1980–2015[J]. Sustainability, 2018, 10(11): 3927.
[56] Dadashpoor H, Azizi P, Moghadasi M. Analyzing spatial patterns, driving forces and predicting future growth scenarios for supporting sustainable urban growth: Evidence from Tabriz metropolitan area, Iran[J]. Sustainable Cities and Society, 2019, 47: 101502.
[57] Abo-El-Wafa H, Yeshitela K, Pauleit S. Exploring the future of rural–urban connections in sub-Saharan Africa: modelling urban expansion and its impact on food production in the Addis Ababa region[J]. Geografisk Tidsskrift-Danish Journal of Geography, 2017, 117(2): 68-81.
4.Lines 106-110 are typical for the methodology, also figure 1 should in no way be part of the introduction.
Response: Thanks for your comments. We placed the contents lines 106-110 in the original manuscript to to the line 150-151 and 157-159 in our revised version. Besides, we placed the figure 1 to the newly added section: 2.3.1. Technical flowchart, and added some contents on line 176-185 “First, based on the land use data and contribution index (K), the spatial and temporal evolution pattern of cultivated land occupied by urban expansion was analyzed, and the contribution of cultivated land in urban expansion was clarified. Second, from the dimensions of social elements, economic elements, traffic elements, and policy elements, the evaluation index system of cultivated land occupied by urban expansion was constructed. Third, the screened factors were input into the GWR model and OLS model to obtain the model with the best fitting effect. Last, based on the regression results of the optimal model, the driving force types and evolution paths of cultivated land occupied by urban expansion were analyzed. The technical framework is shown in Figure 2.”
- Figure 2: There is no research area location for at least the entire area ofChina
Response: Thanks for your comments. We revised the Figure 2, and showed the location of the study area in China.
- The term 'contribution index' is not clearly defined.
Response: Thanks for your comments. We defined the conception of “contribution index” on line 196-199 in our revised manuscript as “The contribution index (K) [48] refers to the proportion of the occupied cultivated land in the newly added urban construction land area in different periods, which reflects the dependence of urban expansion on cultivated land occupation and the contribution of cultivated land to urban expansion.”
7.The content and interpretation of the data in Table 2 requires clarification. What does "Title 1 Index" mean? How should the value in the column "Newly added area" (-185.927 km2) be interpreted, was nearly 200 km2 of built-up areas removed during this period and then restoration of agricultural crops in this area (this is obviously not possible)?
Response: Thanks for your comments. We are sorry for that we did not express the relevant concepts clearly. After inspection, we have revised the "Title 1 Index" to "Period". In original Table 2, “Occupied area” was referred to the occupied cultivated land area in the newly added urban construction land; “Newly added area” was referred to the increase in the cultivated land area occupied by urban expansion in this period compared to the previous period. We just want to use this value to reflect the growth trend of the cultivated land cultivated land occupied by urban expansion in different periods compared with the previous period. “Newly added area” (-185.927 km2) means that the cultivated land area occupied by urban expansion decreased by 185.927km2 in 2010-2015 compared with that of 2005-2010.
The K referred to “the proportion of the occupied cultivated land in the newly added urban construction land area in different periods, which reflect the dependence of urban expansion on cultivated land occupation and the contribution of cultivated land to urban expansion.” In fact, K was calculated based on Equation 1 using “Newly added urban construction land area” and “Occupied cultivated land area in the newly added urban construction land ”. Therefore, the K value is independent of the “Newly added area” in the original manuscript. To avoid confusion, we removed the “Newly added area” column. In addition, we apologize for not adding the necessary information for the “Newly added urban construction land area” due to our negligence. Therefore, we revised the Table 2 in page 7 in the revised manuscript as follows:
Table 2. Statistics on urban expansion dynamics occupying cultivated land from 2000-2018.
|
Period |
Newly added urban construction land area(km²) |
Occupied cultivated land area in the newly added urban construction land (km²) |
K |
|
2000-2005 |
184.550 |
172.057 |
93.23 |
|
2005-2010 |
233.204 |
206.916 |
88.73 |
|
2010-2015 |
26.828 |
20.989 |
78.24 |
|
2015-2018 |
188.256 |
70.566 |
37.48 |
8.What do the authors of the article understand by "cultivated land occupied by urban construction land"? do the authors show the intended use of the areas for future construction investments (spatial planning stage), or is the area actually already built-up?
Response: Thanks for your comments. The "cultivated land occupied by urban construction land” means that: with the development of urbanization, the cultivated land area occupied in the new added construction land in different periods. In fact, it represents the area actually already built-up.
9.Is the value of the K-factor in table 2 correctly calculated? I don't think so (looking at the formula provided by the authors)
Response: Thank you for your meticulous review. We are so sorry for that. After inspection, we have corrected the value of K in Table 2 as follows:
Table 2. Statistics on urban expansion dynamics occupying cultivated land from 2000-2018.
|
Period |
Newly added urban construction land area(km²) |
Occupied cultivated land area in the newly added urban construction land (km²) |
K |
|
2000-2005 |
184.550 |
172.057 |
93.23 |
|
2005-2010 |
233.204 |
206.916 |
88.73 |
|
2010-2015 |
26.828 |
20.989 |
78.24 |
|
2015-2018 |
188.256 |
70.566 |
37.48 |
- The remaining elements of the presented results and their discussion seem to be flawless. However, again, as with the introduction, the discussion lacks any reference to the results obtained and to existing results from other parts of the world.
Response: Thanks for your comments. in the Conclusion, relevant comments and research comparisons are added, and we also added relevant literatures. For example, line 423-436, we added “Cultivated land protection has become a major factor influencing sustainable social and economic development [49],[50]. Some countries, such as China, United States, Japan, etc., have implemented strict cultivated land protection policies to control the rapid conversion of cultivated land into urban construction land. In fact, there have been discussions and disputes about the cultivated land occupied by urban expansion. Some scholars believe that the cultivated land occupied by urban expansion is the inevitable result of urbanization, which is helpful to avoid the possible loss of environment and agricultural ecosystem, and the decline of agricultural production can be compensated by using modern technologies [51],[52]. In some scholars’ view, the occupation of cultivated land will cause the loss of agricultural ecosystems, affect agricultural production, and threatened food security and farmers ' livelihood strategies [53]. Thus, it should be limited urgently. Some scholars have realized that although urbanization is an inevitable trend of social and economic development, it is necessary to protect cultivated land in the process of urban expansion [54],[55].”; line 456-464 we added “The occupation area of cultivated land and the K value decreased gradually, indicating that the contribution of cultivated land in urban expansion was decreasing. This is consistent with the work of Liu et al. [19]. However, Erasu and Lika, in their study of Ethiopia, found that the area of cultivated land occupied by urban expansion continued to increase, and the contribution of cropland to urban expansion also showed an increasing trend [38]. The reason why their results are inconsistent with this study is may be that the Chinese government has implemented strict cultivated land protection and balance policies, the cultivated land area occupied by urban expansion has been effectively controlled.”; line 467-472 “This was consistent with previous research results of Zhong et al. [46], Guan et al. [47] and Dadashpoor et al. [56]. In view of the evolution types and evolution paths of driving forces in terms of urban expansion occupying cultivated land, Chengdu metropolitan area presented diversified spatial development features. Improving the capacity of urban planning is essential to reduce the negative effect of the urban expansion and ensure the food security [57]. ”
The added reference as follows:
[49] Skinner, M W.; Kuhn, R G.; Joseph, A E. Agricultural land protection in China: a case study of local governance in Zhejiang Province. Land Use Pol. 2011, 18, 329 340.
[50] Osumanu I K, Ayamdoo E A. Has the growth of cities in Ghana anything to do with reduction in farm size and food production in peri-urban areas? A study of Bolgatanga Municipality[J]. Land Use Policy, 2022, 112: 105843.
[51] Nguyen, T. T., Hegedűs, G., & Nguyen, T. L. Effect of land acquisition and compensation on the livelihoods of people in Quang Ninh district, Quang Binh Province: Labor and income. Land, 2019, 8(6), 91.
[52] Dadi D, Stellmacher T, Azadi H, et al. The impact of industrialization on land use and livelihoods in Ethiopia: Agricultural land conversion around Gelan and Dukem town, Oromia region[J]. Socio-Ecological Change in Rural Ethiopia: Understanding Local Dynamics in Environmental Planning and Natural Resource Management, 2018: 37-59.
[53] Azadi H, Van Acker V, Zarafshani K, et al. Food systems: New‐Ruralism versus New‐Urbanism[J]. Journal of the Science of Food and Agriculture, 2012, 92(11): 2224-2226.
[54] Epp S, Wan X, Singer R, et al. Farmland Preservation and Urban Expansion: Case Study of Southern Ontario, Canada[J]. Frontiers in Sustainable Food Systems, ,42.
[55] Hu Y, Kong X, Zheng J, et al. Urban expansion and farmland loss in Beijing during 1980–2015[J]. Sustainability, 2018, 10(11): 3927.
[56] Dadashpoor H, Azizi P, Moghadasi M. Analyzing spatial patterns, driving forces and predicting future growth scenarios for supporting sustainable urban growth: Evidence from Tabriz metropolitan area, Iran[J]. Sustainable Cities and Society, 2019, 47: 101502.
[57] Abo-El-Wafa H, Yeshitela K, Pauleit S. Exploring the future of rural–urban connections in sub-Saharan Africa: modelling urban expansion and its impact on food production in the Addis Ababa region[J]. Geografisk Tidsskrift-Danish Journal of Geography, 2017, 117(2): 68-81.
Reviewer 3 Report
Thank you for giving me this opportunity to read the manuscript entitled "Spatio-temporal heterogeneity and driving force evolution of cultivated land occupied by urban expansion in the Chengdu metropolitan area". The topic of this manuscript is interesting. However, some issues still need to be addressed before it could be considered for publication in Land.
1. Please replace the keywords that already appear in the manuscript's title with close synonyms or other keywords, which will also facilitate your paper to be searched by potential readers. Besides, the keywords including quite some words are not suitable to be used as keywords, for example “cultivated land occupied by urban expansion”.
2. The resolution of the images needs to be increased appropriately so that the reader can clearly read the text information.
3. The algorithm used to generate the spatial weight matrix needs to be specified.
4. The finalized bandwidth(s) used in GWR model(s) needs to be shown in the results section.
5. In addition to spatial heterogeneity, temporal heterogeneity has also been a concern in making long time series studies. The authors do not need to use the GTWR model for reanalysis, but the uncertainty arising from ignoring temporal heterogeneity can be discussed in the Limitation section.
6. The resolution of the images used in Figures needs to be increased appropriately so that the reader can clearly read the text information.
7. I believe the Equation (2) in page 6 should be Equation (3).
8. Line 53-54 “On the other hand, it degraded ecosystem service functions and had an important effect on the ecological environment and food security [8].” ].": Some newly published papers focusing on these issues could be cited to support the statement here, for example, the paper titled "How does urban expansion impact people's exposure to green environments? A comparative study of 290 Chinese cities.".
9. Some grammatical errors exist in the manuscript. Therefore, a critical review of the manuscript language will improve readability.
Author Response
Dear reviewer,
Thanks for your comments concerning our manuscript entitled “Spatio-temporal heterogeneity and driving force evolution of cultivated land occupied by urban expansion in the Chengdu metropolitan area” (Manuscript ID: land-1873786). Those comments are very valuable and helpful for revising and improving our paper, as well as the important guiding significance to our researches. I have studied your comments carefully and revised our manuscript which we hope meet with approval. Revised contents are marked in red in the our manuscript. The responses to your comments are as following:
- Please replace the keywords that already appear in the manuscript's title with close synonyms or other keywords, which will also facilitate your paper to be searched by potential readers. Besides, the keywords including quite some words are not suitable to be used as keywords, for example “cultivated land occupied by urban expansion”.
Response: Thanks for your comments. We have revised the keywords. For example, “cultivated land occupied by urban expansion” was revised to “urban expansion” and “cultivated land occupation”; “driving force evolution path” was revised to “driving force” and “evolution path” . Besides, we deleted “food security”.
- Theresolution of the imagesneeds to be increased appropriately so that the reader can clearly read the text information.
Response: Thanks for your comments. We improved resolution of the images, and replaced the original images in our revised manuscript.
- Thealgorithm used to generate the spatial weight matrixneeds to be specified.
Response: Thanks for your comments. We selected the Gauss function as the spatial weight function to generate the spatial weight matrix. We specified the algorithm used to generate the spatial weight matrix, and add the formula on line 225-230 in the revised manuscript.
- The finalized bandwidth(s) used in GWR model(s) needs to be shown in the results section.
Response: Thanks for your comments. We added the finalized bandwidth(s) used in GWR model(s) in the results section on line 303-304 in the revised manuscript as follows “the optimal fixed bandwidth value was 2214450.498.”.
- In addition to spatial heterogeneity, temporal heterogeneity has also been a concern in making long time series studies. The authors do not need to use the GTWR model for reanalysis, but the uncertainty arising from ignoring temporal heterogeneitycan be discussed in the Limitation section.
Response: Thanks for your comments. We added the discussion about the ignoring temporal heterogeneity in the Limitation section on line 493-498 in our revised manuscript as follows: “this paper only studied the spatial heterogeneity of driving forces of cultivated land occupied by urban expansion, without considering the uncertainty caused by the temporal heterogeneity simultaneously. In the future, GTWR model which can take into account the temporal and spatial heterogeneity of driving factors will be used to analyze the driving forces of cultivated land occupied by urban expansion.”. As you pointed that our manuscript is weakness of temporal heterogeneity, we changed the title of our manuscript to “Spatio-temporal pattern and driving force evolution of cultivated land occupied by urban expansion in the Chengdu metropolitan area”.
- The resolution of the images used in Figures needs to be increased appropriately so that the reader can clearly read the text information.
Response: Thanks for your comments. We improved resolution of the images, and replaced the original images in our revised manuscript.
- I believe the Equation (2) in page 6 should be Equation (3).
Response: Thanks for your comments. We revised the “ Equation (2)” to “Equation (3)”.
- Line 53-54 “On the other hand, it degraded ecosystem service functions and had an important effect on the ecological environment and food security [8].” ].": Some newly published papers focusing on these issues could be cited to support the statement here, for example, the paper titled "How does urban expansion impact people's exposure to green environments? A comparative study of 290 Chinese cities.".
Response: Thanks for your comments. We added this literature into our revised manuscript on line 565-566.
- Some grammatical errors existin the manuscript. Therefore, a critical review of the manuscript language will improve readability.
Response: Thanks for your comments. Some grammatical errors were corrected in our revised manuscript. For example, on line 20-23, the original sentence revised to “the total area of cultivated land occupied by urban expansion from 2000–2018 was 632.838 km2, and mainly concentrated in the main urban area of Chengdu City and its surrounding districts and counties.”; line 53, “is ” revised to “are”; line 196-199, the concept of K was revised to “The contribution index (K) [48] refers to the proportion of the occupied cultivated land in the newly added urban construction land area in different periods, which reflects the dependence of urban expansion on cultivated land occupation and the contribution of cultivated land to urban expansion”; line 201-202, the concept of was revised to “and is the occupied cultivated land area in the newly added urban construction land. ”; relevant descriptions in the Table 2 were revised; line 449, the “ determining ” was revised to “determine”, etc.
Round 2
Reviewer 2 Report
I have read the responses to the comments in the review, as well as the revised version of the article. The new version of the manuscript has been significantly improved as a result of taking into account the comments contained in individual reviews. The current version is in my opinion suitable for publication.
Reviewer 3 Report
Thank you for giving me this opportunity to read the revised version of the manuscript titled "Spatio-temporal heterogeneity and driving force evolution of cultivated land occupied by urban expansion in the Chengdu metropolitan area", and for the detailed responses to my earlier comments. I am satisfied with this revised version, and I think it is acceptable now.